# Autologous Micro-Fragmented Adipose Tissue (MFAT) to Treat Symptomatic Knee Osteoarthritis: Early Outcomes of a Consecutive Case Series

**DOI:** 10.3390/jcm10112231

**Published:** 2021-05-21

**Authors:** Wouter Van Genechten, Kristien Vuylsteke, Pedro Rojas Martinez, Linus Swinnen, Kristof Sas, Peter Verdonk

**Affiliations:** 1MoRe Institute, 2100 Antwerp, Belgium; kristien.vuylsteke@azmonica.be (K.V.); pverdonk@yahoo.com (P.V.); 2Department of Orthopaedic Surgery, Antwerp University, 2000 Antwerp, Belgium; 3Cirugía Ortopédica, Hospitales Ángeles, 72190 Puebla, Mexico; pedrorojma@hotmail.com; 4Department of Radiology, AZ Monica, 2100 Antwerp, Belgium; linus.swinnen@icloud.com; 5Orthopaedic Center (ORTHOCA), 2000 Antwerp, Belgium; kristof_sas@msn.com

**Keywords:** knee, osteoarthritis, biological, adipose tissue, autologous

## Abstract

The study aimed to evaluate the short-term clinical effect, therapeutic response rate (TRR%), and therapy safety of a single intra-articular autologous MFAT injection for symptomatic knee OA. Secondly, patient- and pathology-related parameters were investigated to tighten patient selection for MFAT therapy. Sixty-four subjects with symptomatic mild–severe knee OA were enrolled in a single-center trial and received a unilateral (*n* = 37) or bilateral (*n* = 27) MFAT injection. After liposuction, the adipose tissue was mechanically processed with the Lipogem^®^ device, which eventually produced 8–10 cc of MFAT. Subjects were clinically assessed by means of the KOOS, NRS, UCLA, and EQ-5D at baseline and 1, 3, 6, and 12 months after injection. Adverse events were recorded at each follow-up timepoint. The TRR was defined according to the OMERACT-OARSI criteria and baseline MRI was scored following the MOAKS classification. The TRR of the index knee was 64% at 3 months and 45% at 12 months after injection. Therapy responders at 12 months improved with 28.3 ± 11.4 on KOOS pain, while non-responders lost −2.1 ± 11.2 points. All clinical scores, except the UCLA, improved significantly at follow-up compared to baseline (*p* < 0.05). In the bilateral cohort, no difference in baseline scores or TRR was found between the index knee and contralateral knee (n.s.). An inflammatory reaction was reported in 79% of knees and resolved spontaneously within 16.6 ± 13.5 days after MFAT administration. Numerous bone marrow lesions (BML) were negatively correlated with the TRR at 12 months (*p* = 0.003). The study demonstrated an early clinical improvement but a mediocre response rate of 45% at 12 months after a single intra-articular injection with autologous MFAT. Assessment of bone marrow lesions on MRI can be helpful to increase the therapeutic responsiveness of MFAT up to 70% at 12 months. In comparison to repetitive injection therapies such as cortisone, hyaluronic acid, and PRP, administration of MFAT might become a relevant alternative in well-selected patients with symptomatic knee OA.

## 1. Introduction

Osteoarthritis (OA) is the most prevalent joint disease in the United States with 38–47% of people older than 60 years affected by knee OA [1,2]. Over half of the persons suffering from symptomatic knee OA are younger than 65 years and experience significant disabilities on a daily basis [3]. These numbers are only expected to increase due to an aging population and regrettable obesity numbers [1,4]. Respecting the current survival rates of primary and revision total knee arthroplasty (TKA) [5], many young patients with symptomatic knee OA are considered ‘premature’ for joint resurfacing surgery. Therefore, minimally invasive conservative therapies that are mainly focused on symptomatic relief seem to be more eligible in this growing patient population.

In therapeutic knee OA management, modifiable risk factors such as weight and activity level, often in combination with OA education, physiotherapy, and administration of oral pain medication are targeted first [2,6]. In patients who remain unresponsive, intra-articular injection therapies such as corticosteroids, hyaluronic acid (HA), platelet-rich plasma (PRP) or mesenchymal stromal cells (MSCs) are considered [2]. The latter option is nowadays thoroughly explored in all facets, while PRP reviews and RCTs with HA are showing contradictory outcomes due to the lack of high-level clinical evidence [2,6]. Moreover, injection therapies in general seem to be largely reserved for chronic low-grade knee OA and if already effective, the therapeutic durability and reproducibility is a major concern [2]. Therefore, the recruitment of (autologous) MSCs to combat OA symptomatology is truly deserving of the necessary research attention in an attempt to expand the clinical toolbox of conservative, minimally invasive OA therapeutics.

For autologous transplantation purposes, bone marrow and adipose tissue are the two predominant sites for MSC harvest. Although MSCs from both origins appear to have a similar differentiation potential, bone marrow harvest is more invasive and the substrate contains a lower relative concentration of MSCs compared to adipose tissue [7,8,9]. Moreover, the viability and differentiation capacity of bone marrow MSCs is affected by age, which seems highly relevant considering the majority of knee OA patients are middle-aged [7].

Meanwhile, subcutaneous fat tissue possess an easily accessible and often abundant stock to harvest MSCs (AMSCs) [7,10]. Minimally invasive lipo-aspiration can be performed under local anesthetics in the outpatient setting [7]. Next, the lipoaspirate can either be processed enzymatically or mechanically, which has recently been found to result in important differences regarding the final product [9,11,12,13]. In micro-fragmented adipose-tissue (MFAT) processing, the lipoaspirate is refined to a cluster of 0.2–0.8 mm while the supportive vascular stromal niche remains intact and MSCs/pericytes stay in their natural habitat [9]. Currently, autologous MFAT therapy processed by Lipogems^®^ (Lipogems International SpA, Milan, Italy) is the only FDA approved system that appears to be safe for administering AMSCs to the knee joint. However, clinical MFAT efficacy in the knee OA patient seems limited, as it is mainly being provided by small volume case series or case reports, often in combination with arthroscopic lavage procedures [14,15,16,17,18,19]. Therefore, this study primarily aimed to evaluate the short-term clinical effect, therapeutic response rate (TRR%), and therapy safety of a single intra-articular autologous MFAT injection for symptomatic knee OA. Secondly, patient- and pathology-related parameters were investigated for their potential predictive value to tighten patient selection.

## 2. Materials and Methods

A single-center, prospective case series was conducted to evaluate autologous MFAT for the indication of symptomatic knee OA. From March 2017 to January 2019, subjects were assessed for study eligibility according to the following inclusion criteria: primary or secondary uni- or bilateral symptomatic knee OA evidenced by radiographs or magnetic resonance imaging (MRI), failure of at least one conservative treatment for 3 months (oral pain medication, physiotherapy, other intra-articular injection therapy, etc.), age 25–80 years and willing to comply with study requirements. Patients with a baseline KOOS pain >72 points, the diagnosis of a systemic inflammatory pathology (e.g., rheumatoid arthritis), a known allergy to local anesthesia, an active malignancy, an active knee joint or skin infection at the site of injection, and a significant disease which might potentially interfere with study participation and outcomes (neurological or psychiatric pathology, alcohol or drug abuse, etc.) were excluded. Ethical study approval was obtained from the respective local institute and academic ethical commission (on 23 October 2017). Study purpose, procedures, and measures were extensively discussed with eligible subjects and written informed consent was obtained before study enrolment. The study was included in the national registry for clinical trials (#B300201733775) and was executed respecting the Helsinki declaration of 1964 and later amendments.

### 2.1. Lipo-Aspiration and MFAT Processing

Subjects were put in supine position for the liposuction procedure which was performed under local anesthesia. First, two symmetrical spots were marked on the lumbar region of the abdomen and consequently injected with local anesthetics (xylocaine 2% with adrenaline 1:200.000). With a 15-blade scalpel, a small incision was made in which a blunt cannula was placed to rinse the subcutaneous adipose tissue with a mixture of lidocaine 2% (22 × 10 cc flacon) and adrenaline (1 mg/mL) dissolved in 3 L saline. At the time the skin became pale and cold, the area was sufficiently anesthetized and in vasoconstriction, allowing for a comfortable lipo-aspiration with limited amounts of blood. The Lipogem^®^ device (Lipogems International SpA, Milan, Italy) was used for MFAT processing with the detailed procedure described by Bianchi et al. [9]. In brief, the device consists of a closed disposable immersion system (225 mL) that progressively reduces the size of fat clusters in a two-step fashion. After liposuction, the lipoaspirate is injected on top through a large filter connected with a plastic column containing stainless steel marbles. Next, the column is gently shaken for 20–30 s after which a tap with sterile saline is opened to mix with the lipoaspirate. The shaking step in alteration with the washing step is repeated 5–6 times until the marbles become clearly visible. The device is then reversed and with a sterile syringe (10 cc) connected to the bottom, the lipoaspirate is pushed upwards through a second (smaller) cutting hexagonal filter into another empty syringe on top of the device. In this way, 3–4 syringes of approximately 12 cc MFAT were obtained per patient. The remaining fluid in the syringes was separated and removed to increase the AMSCs concentration, which yielded a final MFAT product per subject of 1–2 syringes filled with 8–10 cc/syringe. Finally, the refined adipose substrate was aseptically injected under ultra-sound guidance from the lateral side of the index knee with an 18 gauge needle by an experienced interventional radiologist. Simultaneous MFAT injection of the contralateral knee was allowed per study protocol if the presence of OA was evidenced on recent imaging and was separately assessed on clinical follow-up timepoints (bilateral cohort). Subjects were advised to resume activities of daily living as soon as possible, while avoiding complete rest or immobilization of the injected knee(s). The intake of non-steroid anti-inflammatory drugs (NSAIDS) was discouraged the first 14 days after injection.

### 2.2. Clinical and Safety Assessment

Subjects demographics (age, sex, BMI, smoking) were recorded at the screening visit after study enrollment was confirmed. Clinical outcomes were assessed by means of the knee injury and osteoarthritis score (KOOS), numeric rating scale (NRS) for pain, the University of California in Los Angeles (UCLA) score for activity, and the EQ-5D for overall health status at baseline and 1, 3, 6, and 12 months after injection. The OMERACT-OARSI criteria were used to determine the therapeutic response rate (TRR%) at each follow-up with pain change evaluated by the KOOS pain subscale, functional change by the KOOS activities of daily living (ADL) subscale, and global health assessment change by the KOOS quality of life (QoL) subscale (Figure 1) [20]. In addition, the bilateral injected patients were sub-analyzed to compare outcomes of the index and contralateral knee. Duration, type, and severity of adverse events (AEs) were recorded as defined in Table 1 and categorized under liposuction-related, MFAT-related, or others. As part of the inflammatory triad, ‘pain’, ‘effusion’, and ‘stiffness’ of the index knee were considered separate AEs.

### 2.3. Imaging

Imaging of the affected knee(s) included standard X-rays (Antero-posterior (AP), Rosenberg and skyline view) or magnetic resonance imaging (MRI) to confirm the diagnosis of cartilage lesions or knee OA. X-rays were graded for OA according to the Kellgren and Lawrence (K-L) classification while the MRI osteoarthritis knee score (MOAKS) was used to grade (1) cartilage wear over the three knee compartments, (2) bone marrow lesions (BML), (3) patellar synovitis, (4) osteophytes, and (5) meniscal tears with specific attention for posterior horn lesions (PHL) on T1-and T2-weighted fat-saturated coronal, axial, and sagittal MRI images (Table 2) [21]. For medial (MCCW), lateral (LCCW), and the patella-femoral (PFCW) cartilage wear, summation of the scored areas (‘size’) was performed which resulted in a 15-point score for the femorotibial compartments each and in a 12-point score for the PF joint. BML and osteophytes were likewise scored and overall summed to yield a score of respectively 45 points and 36 points. Patellar synovitis was scored as 0–3 points with 0–1 representing minor and 2–3 major synovitis. Meniscal tears were assessed per meniscus part (anterior, body, or posterior) and PHL of the medial (MM) or lateral (LM) meniscus were scored as either present or absent.

Prognostic factors for the outcome after MFAT were assessed in both categorical and regression analysis with distinction for patient- and pathology-related parameters.

### 2.4. Statistics

Regarding descriptive statistics, categorical variables were expressed as frequency/percentage of occurrence while continuous variables were represented as means with standard deviation. Paired-sample *t*-test was used to compare clinical scores between different timepoints. For subgroup categorization analysis, the two-sided independent *t*-test and Fischer’s exact test were applied to compare respectively the means of absolute clinical improvement and the TRR% at any given follow-up timepoint. Because of the large sample size and low number of drop-outs, missing data were handled with a single imputation model under ‘last observation carried forward.’ Multiple regression analysis was performed for the outcomes TRR and absolute clinical improvement (KOOS pain) at all study timepoints. Regression models were fit separately for patient-specific (age, sex, and BMI) and pathology-specific (MCCW, LCCW, PFCW, patellar synovitis, BML, osteophytes, and PHL medial and lateral meniscus) factors. Significance level was set at α = 0.05. All statistics were conducted in JMP 15.0 (Cary, NC, USA).

## 3. Results

Sixty-four subjects were included in the study and the index knee was treated with a single injection of autologous MFAT prepared with the Lipogem^®^ device (Figure 2). In 27 subjects, the contralateral knee was also symptomatic and simultaneously injected with MFAT (bilateral cohort). Subject demographics are outlined in Table 3. Pre-injection MRI was available in 73% (47/64) and scored by the MOAKS classification. Other subjects (*n* = 17) had an X-ray of the index knee which was scored as K-L grade 1 (*n* = 1), grade 2 (*n* = 5), grade 3 (*n* = 7), and grade 4 (*n* = 4). One subject (2 knees) was lost to follow-up after 3 months of injection. Between 6–12 months, 10 knees (11%) (8 index knees and 2 contralateral knees) required surgical intervention in the form of TKA (8/10), patellar release (1/10), or patellofemoral arthroplasty (1/10). One patient (2 knees) withdrew informed consent at 6 months.

### 3.1. Safety 

A total of 119 adverse events in 53 subjects occurred over 12 months of study follow-up (unilateral and bilateral injections; Appendix A). At least one parameter of the inflammatory triad (pain, swelling, or stiffness) was found in 72 injected knees (79%) as a consequence of MFAT. The inflammatory reaction resolved spontaneously after 16.6 ± 13.5 days and was predominantly moderate in character (52%). No serious adverse events related to MFAT were observed. The TRR in subjects with an initial reaction at 1 month was 35% versus 53% in knees with complete absence of an inflammatory reaction. At 12 months follow-up, the TRR was respectively 51% and 21% (*p* = 0.07). Three subjects had an AE that was possibly related to MFAT in the form of subjective knee instability (2 subjects) and muscle aching in the calves (1 subject). Finally, 5 subjects experienced an AE that was neither MFAT- nor liposuction-related; gallstones (1 subject), stroke (2 subjects), and tendinopathy (3 subjects).

### 3.2. Clinical Outcomes Index Knee

All KOOS subscales improved significantly from baseline up to 12 months follow-up (*p* < 0.01) (Figure 3). At 3 months post-injection, all KOOS subscale scores were significantly higher than at 1 month follow-up (*p* < 0.01) and thereafter slightly decreased up to 12 months. The overall TRR was 64% at 3 months and 45% at 12 months follow-up (Figure 4). Therapy responders at these timepoints improved 28.3 ± 13.0 points and 28.3 ± 11.4 on KOOS pain, respectively. Therapy non-responders lost −1.9 ± 9.7 points at 3 months and −2.1 ± 11.2 points at 12 months on KOOS pain relative to baseline. The NRS score for pain overall decreased from 5.5 ± 2.2 to 3.8 ± 2.4 at 3 months and to 4.2 ± 2.5 at 12 months (*p* < 0.01). Finally, no significant difference in the UCLA activity level was observed in both the unilateral and the bilateral injected cohort. The EQ-5D index improved significantly in the unilateral cohort at 3, 6, and 12 months relative to baseline (Appendix A).

### 3.3. Subgroup Analysis 

#### 3.3.1. Bilateral Cohort 

Twenty-seven study subjects (age 55.6, 52% females and BMI 28.2) received a simultaneous MFAT injection in both the index knee and the contralateral knee, which was derived from the same MFAT preparation device. At 3 months, the TRR was 59% for the index knee and 67% for the contralateral knee (n.s.). At 12 months follow-up, a TRR of 48% was found for both the index and the contralateral injected knee.

#### 3.3.2. Subject- and Pathology-Specific Factors and Clinical Outcome

Subject’s age, sex, and BMI were investigated in both a regression and categorization model for their potential effect on baseline score and 3 or 12 month clinical benefit (absolute KOOS pain improvement and TRR) after MFAT. Age was an indicator at 3 months follow-up with patients ≤50 years found to have a higher TRR (83%) compared to older patients (53%) (*p* = 0.0161). Sex was an independent predictor for baseline KOOS pain score (*p* = 0.012) and was confirmed by a significantly lower baseline KOOS pain in females (42.8 ± 12.5) compared to males (49.3 ± 10.7) (*p* = 0.0292). Other patient-specific factors were not found to have predictive value. MOAKS outcomes showed a heterogeneous OA population based on cartilage wear, synovitis, osteophytes, BML and meniscus lesions in the posterior horn (Figure 5 and Table 4). The regression analysis for MRI parameters revealed a negative impact of BML on the TRR (−0.17, *p* = 0.003) and absolute KOOS improvement (−1.36, *p* = 0.033) at 12 months follow-up. BML categorization based on severity demonstrated that subjects with no-mild BML were having a significantly higher TRR compared to severe BML subjects at respectively 3 months (92% vs. 47%, *p* = 0.0157) and 6 months (77% vs. 33%, *p* = 0.0296) post-injection (Figure 6). Finally, the TRR for PHL of the medial meniscus significantly differed at 6 months in favor of subjects without PHL (71% vs. 26%, *p* = 0.003).

## 4. Discussion

The most important findings of the study are a TRR of 64% at 3 months and 45% at 12 months with therapy responder improvement of 28.3 ± 13.0 and 28.3 ± 11.4 points on KOOS pain, respectively. Interestingly, a subgroup of patients with only mild BML on baseline MRI (28%) was identified, showing a TRR of 77% at 6 months and 69% at 12 months follow-up. During the first 2–4 weeks after MFAT administration, an inflammatory reaction (flare) was observed as shown by the presence of at least one inflammatory sign in 79% of injected knees. The majority of flares appeared to resolve spontaneously within the first month under oral NSAID prohibition and early mobilization during the first 2 weeks. At 12 months follow-up, a trend towards a higher TRR (*p* = 0.07) was found for subjects with initial flares after MFAT (TRR = 51%) compared to non-flare knees (TRR = 26%). This suggests that an early inflammatory reaction after MFAT might indicate a more sustainable effect and did not negatively affect the 12 month clinical outcome in this study.

For other intra-articular injection therapies such as cortisone, an inflammatory flare can be expected in 2–25% within the first hours after administration [2]. Although corticosteroid flares may only persist for 2–3 days, they similarly do not tend to worsen the therapeutic outcome [2]. Regarding HA injections, an increased risk of flares and granulomatous inflammation was earlier established [22]. Considering PRP injections, an inflammatory reaction of mild–moderate character is commonly observed in the majority of patients (>80%), while in our series, subjects had mostly reactions of moderate intensity [23]. The effect of initial flares on 6 and 12 month therapeutic outcomes, however, is generally underreported in most studies and might be an important early indicator for effectivity of autologous biological treatments. Smaller clinical MFAT studies for knee OA barely observed inflammatory reactions after percutaneous intra-articular administration in the past, which contrasts with our results [16,18,24]. Nevertheless, the intra-articular injection of AMSCs for OA has been repeatedly reported safe and well-tolerated with minimal risk of infection, malignancy development, or severe clinical deterioration [25,26,27]. As suggested by Barfod et al. [18], patients eligible for MFAT should be properly informed about abdominal discomfort after lipo-aspiration and significant flares (pain, swelling, or stiffness) during the first 2–4 weeks, as found in this study. Of note, arthralgia after MFAT injection in this study was assumed to be part of an inflammatory reaction, although it could also be attributed to a mechanical volume effect of MFAT (8–10 cc).

Age was a mediocre predictor for the early TRR after MFAT. Patients ≤50 years were found to have a higher TRR at 3 months (83%) compared to older patients (53%) (*p* = 0.0161). However, this difference was nullified at 12 months post-injection. Age limitation for MFAT application seems therefore undetermined. The baseline clinical status of our patient cohort did not seem to differ much compared to similar recent studies investigating the use of biologicals in knee OA [28,29,30]. In addition, female sex was an actual determining factor for lower baseline KOOS pain scores (*p* = 0.014), which supports the paradigm that women are in general more severely affected by knee OA compared to men [31].

Considering the baseline MRI parameters, a negative association of BML with the TRR and absolute pain improvement at 12 months after injection was demonstrated. Additionally, by categorizing BML for severity and quantity, patients with mild BML were having a significantly higher TRR at 3 (92%) and 6 months (77%) compared to patients with severe BML (resp. 47% and 33%). It is well-known that lesions in the subchondral bone can be a primary cause of pain in the arthritic knee joint [32]. Since these lesions are located extra-articularly, it is unsurprising that subjects with numerous BML are poor responders after 6 months to an intra-articular injection therapy such as MFAT. In general, these results indicate that knee OA patients with severe BML seem to be less ideal candidates for MFAT. Importantly, the severity of baseline cartilage wear (in any compartment) and synovitis status were not found to determine the outcome after MFAT administration. As a limiting factor, not all subjects received a pre-injection MRI but it was available in a reasonable number of index knees (73%, *n* = 47) to perform regression analysis.

Effect durability of conservative treatments for knee OA forms an important criterion in considering its application. In regards to established injection therapies, corticosteroids have a clinical benefit range of 8–66 days with a generally rapid onset of action [2]. Despite common application in clinical practice, inconsistent efficacy and durability results have been reported for HA injections, which are typically administered 2–3 times within one month [2,22]. However, recent OARSI guidelines (2019) conditionally advise HA infiltrations because of proven pain relief at 12 weeks or beyond and state that strong evidence is lacking for PRP [6]. Autologous PRP is currently the most clinically applied biological therapy for knee OA and has shown its benefits over HA injections in the younger patient (<50 years) with mild–moderate OA [33,34,35]. Filardo et al. showed a median effect duration of 9 months after triple PRP injections with a 3-week interval [36]. Very similar to our results with MFAT, Campbell et al. concluded in a systematic review that effect onset of PRP is starting 2 months after administration and potentially might last for 12 months post-injection [35]. Multiple PRP injections however were associated with an increased risk of local adverse events. For single autologous protein solution (APS) injections, Kon et al. demonstrated a progressively increasing TRR according to the OMERACT-OARSI criteria of 50% at 3 months and 65.5% at 12 months in patients with moderate knee OA (KL grade 2–3) [28]. Moreover, they were able to show a relevant difference in the WOMAC pain score with saline control at 12 months follow-up, but not for the TRRs of both therapies. A single MFAT injection in this study demonstrated an TRR of 56% at 6 months and 45% at 12 months, which is fairly low compared to other bio-injectables and saline controls [28]. However, when only considering subjects with mild BML (28%) on MRI, a TRR of 77% at 6 months and 69% at 12 months was observed. This highlights the importance of patient stratification and the potential value of a pre-injection MRI for designated conservative injection therapies and the individual estimation ratio for success.

Considering the results of the largest published MFAT series to date (*n* = 110 knees), patient reported outcome measures VAS, OKS, and EQ-5D score all improved in statistically significant ways at 12 months follow-up [37]. However, in contrast to our study, no adverse events were reported. The longest clinical follow-up after MFAT administration was published by Boric and Hudetz (*n* = 32 knees) and showed significantly lower VAS scores at 12 and 24 months [17,24]. Interestingly, they found a relevant improvement in cartilage glycosaminoglycan (GAG) content on MRI (dGEMRIC analysis) at both timepoints, opposing the natural GAG decrease associated with OA. Future research endeavors are needed to bring more clearance on the clinical durability and indication of a single MFAT injection for knee OA; however, the value, safety, and timing of multiple MFAT injections remains unknown and should be determined by large, well-designed studies.

Finally, this study was not conducted without limitations. Twenty-seven subjects were injected bilaterally which might have compromised clinical outcome interpretation. At follow-up timepoints, subjects of the bilateral cohort were closely monitored and it was repetitively emphasized that each injected knee joint had to be scored individually (KOOS and NRS). For the UCLA and EQ-5D index scores, knee-specific evaluation was impossible and outcomes were therefore presented for the unilateral- and bilateral-injected cohort separately (Appendix A). Next, oral dose-specific analgesics during the study course were not recorded, which might have interfered with pain and functional outcomes after MFAT. The use of NSAIDS was strongly discouraged the first 2 weeks post-injection but the administration of acetaminophen and low-dose tramadol were allowed per physician’s prescription. However, since these types of pain medication were often considered in an earlier OA stage (i.e., past OA treatments that failed to provide sufficient pain relief), their contribution to the clinical effect after MFAT was assumed to be of minor significance. The age range for study inclusion was fairly large (25–80 years). Except for two patients with the respective ages of 27 and 35 years (post-traumatic OA), all subjects (97%) were between 40 and 75 years old. Lastly, this study did not include a control group since it had merely observational intentions to assess MFAT as a potential alternative for managing symptoms in a heterogeneous knee OA population. Nevertheless, a randomized controlled trail is ongoing which aims to evaluate direct comparison of MFAT with hyaluronic acid infiltrations.

## 5. Conclusions

The study demonstrated an early clinical improvement but a mediocre response rate of 45% at 12 months after a single intra-articular injection with autologous MFAT. Assessment of bone marrow lesions on MRI can be helpful to increase the therapeutic responsiveness of MFAT up to 70% at 12 months. A post-injection inflammatory reaction of 2–4 weeks must generally be considered without impairing clinical outcomes. In comparison to repetitive injection therapies such as cortisone, hyaluronic acid, and PRP, administration of MFAT might become a relevant alternative in well-selected patients with symptomatic knee OA.

## Figures and Tables

**Figure 1 jcm-10-02231-f001:**
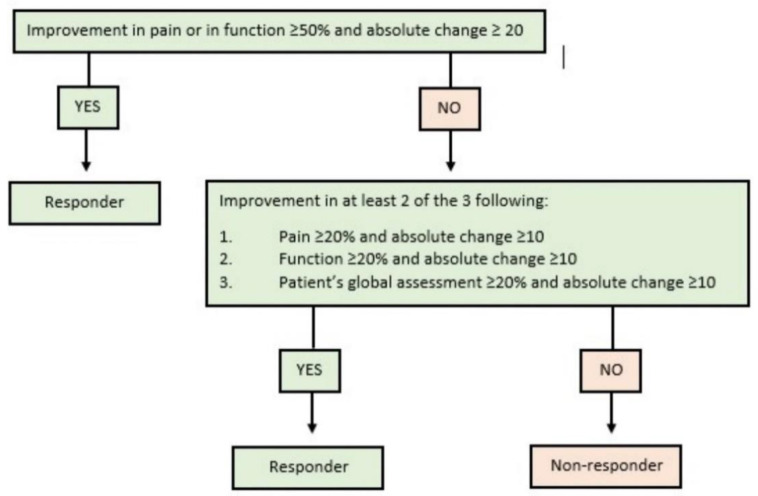
The OMERACT-OARSI criteria to define responding and non-responding subjects to treatment interventions in knee osteoarthritis. The therapeutic response rate (TRR%) is derived from this classification at respective follow-up timepoints after MFAT administration.

**Figure 2 jcm-10-02231-f002:**
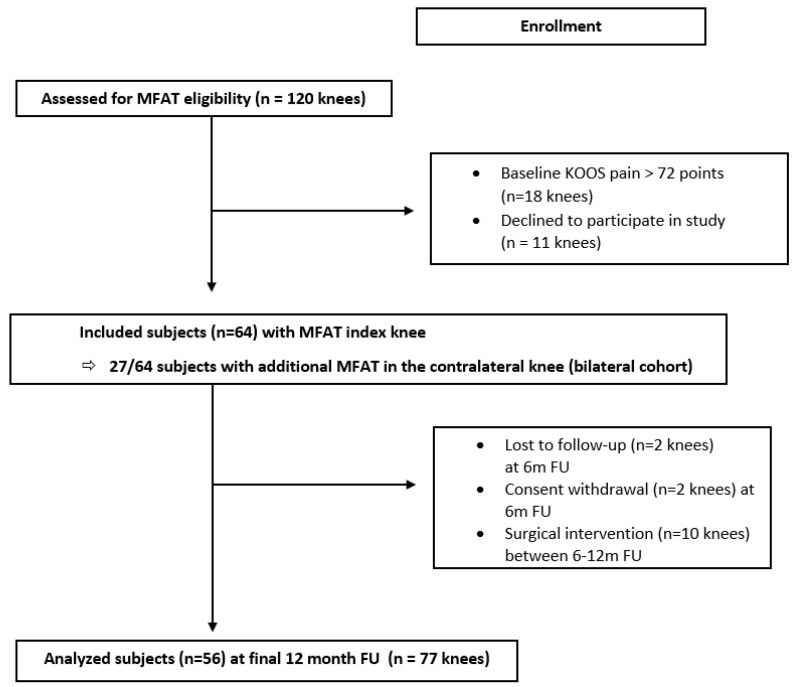
Flowchart of patient enrollment according to the standards of reporting trials statement with numbers of excluded and analyzed subjects. (MFAT, micro-fragmented adipose tissue; FU, follow-up).

**Figure 3 jcm-10-02231-f003:**
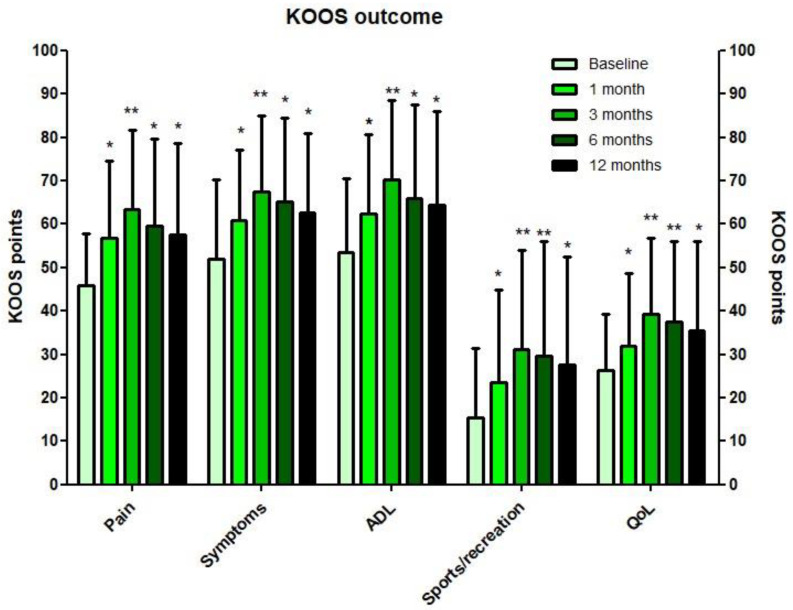
KOOS outcomes of the index knees (*n* = 64) after a single injection with autologous MFAT. (ADL, activities of daily living; QoL, quality of living). * significant difference compared to baseline. ** significant difference compared to baseline and 1 month.

**Figure 4 jcm-10-02231-f004:**
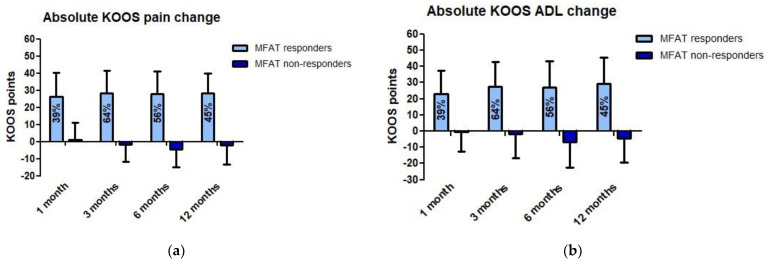
Overall therapeutic response rate (TRR) and associated absolute change in (**a**) KOOS pain and (**b**) KOOS activities of daily living (ADL) relative to baseline.

**Figure 5 jcm-10-02231-f005:**
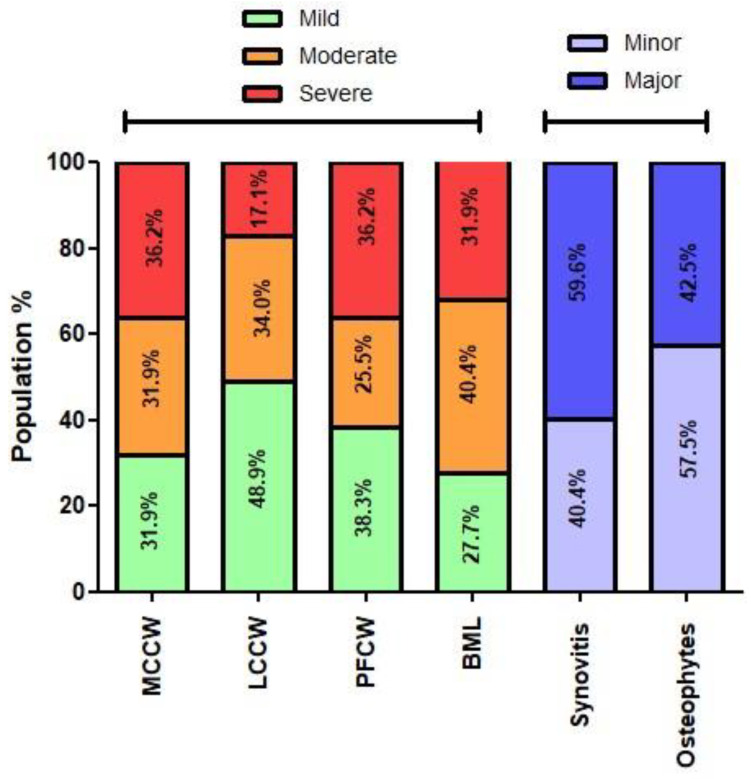
MOAKS outcome on baseline index knee MRI (*n* = 47). (MCCW, medial compartment cartilage wear; LCCW, lateral compartment cartilage wear; PFCW, patella-femoral cartilage wear; BML, bone marrow lesions).

**Figure 6 jcm-10-02231-f006:**
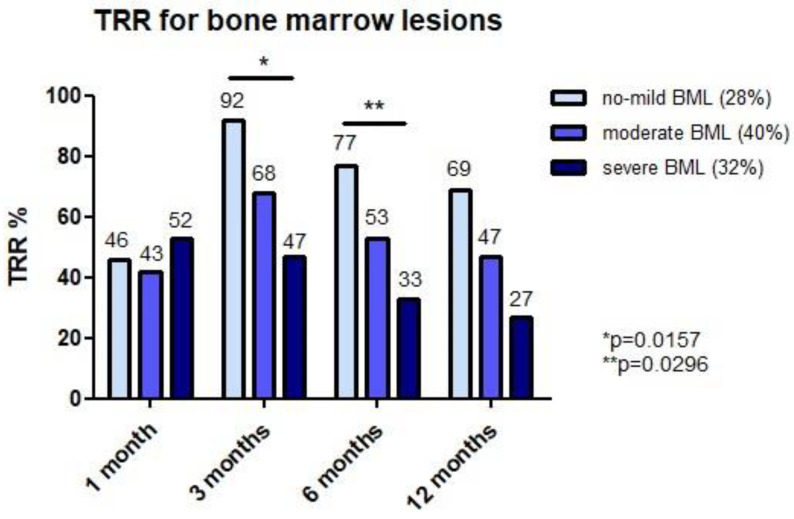
Therapeutic response rate (TRR%) for bone marrow lesion (BML) classification.

**Table 1 jcm-10-02231-t001:** Definitions of adverse event (AE) severity with required interventions.

Severity	Definition	Additional Medication Required?	Physicians Advice Required?
Mild	Minor discomfort noticed but does not interfere with normal daily activity	No	No
Moderate	Discomfort reducing or affecting normal daily activity	Yes	Potentially
Severe	Incapacitating with inability to work or perform normal daily activity	Yes	Yes
Serious	Permanent damage, life-threatening or death	Yes	Yes

**Table 2 jcm-10-02231-t002:** Baseline MRI features of the study population assessed by the MRI osteoarthritis knee score (MOAKS) system.

Pathology-Specific Parameters (Maximum Points on MOAKS)	Gradation (Points on MOAKS)
MCCW (15)	Mild (0–5) Moderate (6–10) Severe (11–15)
LCCW (15)	Mild (0–5) Moderate (6–10) Severe (11–15)
PFCW (12)	Mild (0–6) Moderate (7–9) Severe (10–12)
BML (45)	Mild (0–3) Moderate (4–10) Severe (>10)
Patellar synovitis (3)	Minor (0–1) Major (2–3)
Osteophytes (36)	Minor (0–14) Major (15–36)
PHL (yes/no)	MM LM

MCCW, medial compartment cartilage wear; LCCW, lateral compartment cartilage wear; PFCW, patella-femoral cartilage wear; BML, bone marrow lesions; PHL, posterior horn lesion; MM, medial meniscus; LM, lateral meniscus.

**Table 3 jcm-10-02231-t003:** (1) Basic subject characteristics with clinical assessment at baseline of the index knee. (2) Subject demographics of the bilateral injected cohort.

(1)		
**All MFAT Subjects (*n* = 64)**		
Age (mean ± SD)	54.2 ± 9.1	
Sex (%female)	51.6%	
BMI (mean ± SD)	27.2 ± 4.5	
Bilateral injection n (%)	27 subjects (42%)	
Smoking (%)	12.9%	
Baseline KOOS (mean ± SD)		
Pain	45.9 ± 12.0	
Symptoms	52.0 ± 18.2	
ADL	53.6 ± 16.9	
Sports	15.4 ± 15.9	
QoL	26.3 ± 13.0	
Baseline NRS (mean ± SD)	5.5 ± 2.2	
Baseline UCLA (mean ± SD)	6.1 ± 2.1	
Baseline EQ-5D index (mean ± SD)	0.809 ± 0.052	
(2)		
**Bilateral Cohort (*n* = 27)**	**Index Knee**	**Contralateral Knee**
Age (mean ± SD)	55.6 ± 10.1	
Sex (%female)	51.9%	
BMI (mean ± SD)	28.2 ± 4.5	
Baseline KOOS (mean ± SD)		
Pain	45.2 ± 12.1	46.5 ± 14.1
Symptoms	52.0 ± 19.0	56.6 ± 21.7
ADL	50.2 ± 16.2	53.4 ± 15.3
Sports	14.6 ± 14.8	17.6 ± 16.7
QoL	27.1 ± 12.9	29.9 ± 12.3
Baseline NRS (mean ± SD)	5.4 ± 2.2	4.6 ± 1.9

BMI, body mass index; KOOS, knee injury and osteoarthritis outcome score; ADL, activities of daily living; QoL, quality of living; NRS, numeric rating scale.

**Table 4 jcm-10-02231-t004:** Baseline meniscus status of the index knees on MRI (*n* = 47). Meniscus pathology was scored per part (anterior, body, or posterior part).

Location	Injury	Tear Type
Medial meniscus	Tear	Vertical: 6% (3/47)
		Horizontal: 30% (14/47)
		Complex: 32% (15/47)
		PHL: 40% (19/47)
	Maceration	Partial: 55% (26/47)
		Complete: 32% (15/47)
Lateral meniscus	Tear	Vertical: 11% (5/47)
		Horizontal: 23% (11/47)
		Complex: 13% (6/47)
		PHL: 30% (14/47)
	Maceration	Partial: 45% (21/47)
		Complete: 11% (5/47)

PHL, posterior horn lesion.

## Data Availability

The data presented in this study are available on request from the corresponding author. The data are not publicly available due to the privacy policy on patient consent.

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
