# Peer review of "Autologous Micro-Fragmented Adipose Tissue (MFAT) to Treat Symptomatic Knee Osteoarthritis: Early Outcomes of a Consecutive Case Series"

_jcm, 2021, doi:10.3390/jcm10112231_

Round 1

Reviewer 1 Report

Dear Authors, 

The subject of the article is very interesting and up-to-date.

There is a lack of particularly important information: the degree of degenerative changes, the qualification of patients for this type of treatment, information on prior treatment. 
Moreover, comparing the effects of treatment in patients with the disease on one or both limbs carries a high risk of drawing false conclusions, therefore these 2 groups should not be compared together.

You should improve the methodology to draw proper conclusions.

Author Response

Dear reviewer, 

We would like to thank you for giving your critical view on our manuscript and hope we were able to increase the quality of this work. 

Best regards

Response to reviewer 1

Dear Authors, 

The subject of the article is very interesting and up-to-date.

There is a lack of particularly important information:

  • the degree of degenerative changes

A: We do realize that a general OA scoring on plain radiographs like Kellgren and Lawrence or Albäck score is missing for this study. Nevertheless, since ¾ of the studied index knees had a pre-injection MRI scan, we do think that the MOAKS score is a more reliable tool to asses degenerative changes in each compartment separately, to determine the meniscus status and grade bone marrow edema compared to the OA severity on Xray. Moreover, the outcome of the MOAKS score was used in the regression analysis to link excessive bone marrow oedema to a worse outcome after MFAT infiltration. The detailed outcome of the MOAKS is outlined in figure 5 and shows an heterogenous population of mostly medial and patellofemoral OA ranging from mild to severe damage to the cartilage. Although it seems more complex for interpretation than the commonly used Kellgren and Lawrence classification, we believe that it is much more informative for these specific study purposes.

  • the qualification of patients for this type of treatment

A: Study inclusion and exclusion criteria are outlined on line 89-98. In retrospect, we do think that some criteria such as the age range might raise some concern. The MFAT therapy is novel and to date, few smaller clinical studies have been published on the clinical benefit without focusing on the patient/pathology suitability for MFAT use. Therefore, we decided to inject a broad range of patients (age) and severity of OA in order to identify a patient category for which MFAT might be beneficial. Numerous patients were included as a ‘last resort’ conservative option before decision for arthroplasty was made. The study had merely an observational purpose since no indication for MFAT infiltration has currently identified in clinical practice. Regarding the age range for inclusion, we also included a minority of patients with post-traumatic OA. To give more details about the age of the youngest patients, we included one patient who was 27 and one of 35 years old. Furthermore, all patients were above the age of 40 with the oldest patient being 75. The was clarified and implemented in the discussion section (line 411-413).

As the data are showing, we were able to tighten selection criteria for the MFAT therapy regarding the extension of bone marrow lesion on MRI. Therefore in our current practice, we qualify patients with no to few bone marrow lesions on MRI more for MFAT and explain they will have a higher chance of responding to the therapy (70% at 1 year).

  • Information on prior treatment

A: The authors believe it is hard to meticulously describe the previous conservative treatments of study patients in general, mostly due to practical issues. Often, patients were seen and included coming from another hospital from which previous therapies were not always well documented. As stated in the methodology section (line 91-92), patients could only be included after at least one conservative therapy had failed to bring symptomatic relief over a minimum of 3 months. Although we cannot come up with concrete data for each patient about past treatments because some were treated elsewhere, most of them did physiotherapy and received an intra-articular injection with cortisone or hyaluronic acid without sufficient effect. This was a particularly important criterium for study inclusion since the MFAT infiltration cannot be considered a first line treatment.

Moreover, comparing the effects of treatment in patients with the disease on one or both limbs carries a high risk of drawing false conclusions, therefore these 2 groups should not be compared together.

A: In the manuscript, these two groups were not compared to each other but separate analysis was performed. Bilateral injected patients came into clinics with one symptomatic knee (index knee) and with complains on the contralateral side on a lower level for which injection with MFAT could be justified. It was important to report outcomes separately for those questionnaires that were not ‘knee-specific’ (EQ-5D and UCLA score).

You should improve the methodology to draw proper conclusions.

Reviewer 2 Report

The authors conducted a study focused on the effects of a single injection of micro-fragmented adipose tissue in the management of symptomatic knee OA.

Overall this is a well-designed study, the data is presented in a clear way, the discussion focuses on most of the aspects of the study and the authors clearly admitted the limitations of this paper. I would like to underline that the discussion regarding the effectiveness of various non-operative treatment options is fairly objective and it was a pleasure to read the paper

I believe that the paper will be suitable for publication, however there are several issues that need to be addressed

  1. Description of the harvesting protocol – please use international names of the drugs (line 110- linisol; please state the dose of adrenalin – not ampoules ), also please clarify the statement in line 110, – did you inject 3L of the saline ?
  2. Please provide a clear definition of responders/nonrespnders – what was thecut-off – KOOS levels at baseline or other perhaps a different value ?
  3. Please remove tables 4 and 5 – and attach them as supplementary material; they lengthen the paper and do not provide critical information; this will improve the clarity of the manuscript
  4. Discussion – the authors missed the fact, that they did inject a failry large volume of a non-immediately-resorbable substance into a knee joint – this would most likely be painfull in at least some individuals, yet you consider pain as an inflammatory reaction, perhaps – to some degree at least – this was a “mechanical” raction
  5. I really like that the authors attempted to be objective in their evaluation of the current status of non-operative knee OA management, yet several points should, as I believe, be expanded (perhaps as 1-2 sentences) – as ex. OARSI guidelines provide a different approach
  6. The “discouraging” AAOS guidelines regarding the use of HA seem to be largely ignored by the orthopaedic surgeons in their daily practice, and the OARSI guidelines (https://doi.org/10.1016/j.joca.2019.06.011) state that injections of HA are conditionally recommended.
  7. The authors seem to be in favor of PRP – please discuss with OARSI guidelines
  8. The discussion could benefit from a brief mention of the emerging role of genetics in the treatment of OA (https://doi.org/10.3390/ijms21155430)
  9. The limitations/conclusions – I do understand your enthusiasm for the MFAT injections, but please consider toning down the last paragraphs – a lot of readers could consider your results as “not great but not terrible” and not – as you wrote -“acceptable”. Instead consider rewriting some parts and state that your data shows that his method may indeed be “acceptable” or “promising” or even “providing good outcomes” in carefully selected patients (no MM lesions, no BMLs, younger individuals). Please note – there is an analogy in surgical management of knee OA – there is a strong agreement that implantation of UNIs does result in good functional outcomes ONLY after a careful selection of patients. Perhaps your treatment is like UNIs – it is not an “all-terrain” solution, but has its potential place in the clinical practice.

Author Response

Dear reviewer, 

We would like to thank you for giving your critical view on our manuscript and hope we were able to increase the quality of this work. 

Best regards

Response to reviewer 2

The authors conducted a study focused on the effects of a single injection of micro-fragmented adipose tissue in the management of symptomatic knee OA.

Overall this is a well-designed study, the data is presented in a clear way, the discussion focuses on most of the aspects of the study and the authors clearly admitted the limitations of this paper. I would like to underline that the discussion regarding the effectiveness of various non-operative treatment options is fairly objective and it was a pleasure to read the paper

I believe that the paper will be suitable for publication, however there are several issues that need to be addressed

  1. Description of the harvesting protocol – please use international names of the drugs (line 110- linisol; please state the dose of adrenalin – not ampoules ), also please clarify the statement in line 110, – did you inject 3L of the saline ?

A: International drug names were adjusted in the manuscript (line 109-111). The word ‘delivered’ was changed to ‘rinsed’ as this states more the act of how the patient and subcutis was anesthetized and brought into vasoconstriction with 3L saline. 

  1. Please provide a clear definition of responders/nonrespnders – what was thecut-off – KOOS levels at baseline or other perhaps a different value ?

A: Responder and non-responder criteria were applied as designed by OMERACT-OARSI (Figure 1). We used the KOOS pain improvement (relative to baseline) as for the ‘pain’ criterium, the KOOS activities of daily living (ADL) improvement as for the ‘function’ criterium and the KOOS quality of life (QoL) improvement as for the patient’s global assessment criterium. We think to date, this is the most reliable manner of defining responders of therapeutic interventions in knee osteoarthritis for study purposes. The applied criteria are outlined on line 142-146.

  1. Please remove tables 4 and 5 – and attach them as supplementary material; they lengthen the paper and do not provide critical information; this will improve the clarity of the manuscript.

A: As suggested, table 4 and 5 were removed from the manuscript and attached as supplementary material. We do agree that they take a lot of space. Table numbers were adjusted accordingly.

  1. Discussion – the authors missed the fact, that they did inject a failry large volume of a non-immediately-resorbable substance into a knee joint – this would most likely be painfull in at least some individuals, yet you consider pain as an inflammatory reaction, perhaps – to some degree at least – this was a “mechanical” reaction

A: This is an interesting remark and indeed not specified in the manuscript. Increased knee pain after injection was assumed to be related to inflammatory reaction, but was at least considered as an adverse event. When looking at the adverse event data, 10 patients (mostly bilaterally injected) were suffering from isolated increased knee pain without the association of swelling or stiffness. Therefore the reviewer could be right that there might be a factor of mechanical compression as underlying cause for post-injection arthralgia. This suggestion was implemented and discussed on line 335-337.

  1. I really like that the authors attempted to be objective in their evaluation of the current status of non-operative knee OA management, yet several points should, as I believe, be expanded (perhaps as 1-2 sentences) – as ex. OARSI guidelines provide a different approach

A: the OARSI guidelines were referred in discussion about the use of HA the make it more balanced and to justify its use in practice on specified conditions. Line 360-370.

  1. The “discouraging” AAOS guidelines regarding the use of HA seem to be largely ignored by the orthopaedic surgeons in their daily practice, and the OARSI guidelines (https://doi.org/10.1016/j.joca.2019.06.011) state that injections of HA are conditionally recommended.

A: the OARSI guidelines were referred in discussion about the use of HA the make it more balanced and to justify its use in practice on specified conditions. Line 360-370.

  1. The authors seem to be in favor of PRP – please discuss with OARSI guidelines

A: Again, OARSI guidelines were discussed for PRP use. Line 360-370.

  1. The discussion could benefit from a brief mention of the emerging role of genetics in the treatment of OA (https://doi.org/10.3390/ijms21155430)

A: Although we do believe that genetics might become relevant for patient stratification and proper OA treatment assignment, we do feel that discussing this topic is falling beyond the scope of this work.

  1. The limitations/conclusions – I do understand your enthusiasm for the MFAT injections, but please consider toning down the last paragraphs – a lot of readers could consider your results as “not great but not terrible” and not – as you wrote -“acceptable”. Instead consider rewriting some parts and state that your data shows that his method may indeed be “acceptable” or “promising” or even “providing good outcomes” in carefully selected patients (no MM lesions, no BMLs, younger individuals). Please note – there is an analogy in surgical management of knee OA – there is a strong agreement that implantation of UNIs does result in good functional outcomes ONLY after a careful selection of patients. Perhaps your treatment is like UNIs – it is not an “all-terrain” solution, but has its potential place in the clinical practice.

A: We tried to tone down the interpretation of the results and agree that it’s might become a suitable therapeutic option only for certain individuals (and not for the whole knee OA population). We tried to deliver a primary selection by pointing out the meniscus and bone marrow lesion changes on MRI as a tool to predict the chance of success for the patient (although we do realize this is preliminary). Overall we think we wrote a balanced conclusion stating a mediocre (so not great but not terrible) response rate at 12 months but when patients are well selected, a response rate of 70% at 12 months is possible, which is good. The word ‘acceptable’ was removed.

Reviewer 3 Report

Dear Authors,

as regards the M&M section:

-there is not a OA classification (for example Kellgren-Lawrence)  in order to define exactly the severity of OA of the patients enrolled

-the class age used is to large (from 25 to 80 ys)

-there is not a placebo group in order to verify better the efficacy in the study group

Author Response

Dear reviewer, 

We would like to thank you for giving your critical view on our manuscript and hope we were able to increase the quality of this work. 

Best regards

Response to reviewer 3

Dear Authors,

as regards the M&M section:

-there is not a OA classification (for example Kellgren-Lawrence) in order to define exactly the severity of OA of the patients enrolled

Since ¾ of the studied index knees had a pre-injection MRI scan, we do think that the MOAKS score is a more reliable tool to asses degenerative changes in each compartment, meniscus status and the presence of bone marrow edema compared to the OA grade on Xray. Moreover the outcome of the MOAKS score was used in the regression analysis to link excessive bone marrow oedema to a worse outcome after MFAT infiltration. The detailed outcome of the MOAKS is outlined in figure 5 and shows an heterogenous population of mostly medial and patellofemoral OA ranging from mild to severe damage to the cartilage. Although it seems more complex for interpretation than a commonly used Kellgren and Lawrence classification, we believe it is much more informative for these specific study purposes.

-the class age used is to large (from 25 to 80 ys)

The study had merely an observational purpose since no indication for MFAT infiltration has currently identified in clinical practice. Although not many, we also included few patients with post-traumatic OA. To give more details about the age of the youngest patients, we included one patient who was 27 and one of 35 years old. Furthermore, all patients were above the age of 40 with the oldest patient being 75. The was clarified and implemented in the discussion section (line 411-413).

-there is not a placebo group in order to verify better the efficacy in the study group

As stated in the limitation section (line 413-417), this study had merely an observational purpose of primarily identifying the response rate on MFAT in a broad OA population. We are aware that a control placebo group would have made this study scientifically more powerful and we certainly do not underestimate the effect of placebo (> 50% response on 6 months follow-up). Since only few clinical data are available about MFAT, we think it is more likely to test the therapy in a large population (namely for its safety) before direct comparison. In the meantime, we started a RCT with hyaluronic acid versus MFAT infiltration at our institute to continue on the results of the current study.

Round 2

Reviewer 1 Report

  • bilateral group - there is n.s. difference between limb before treatment- why do You compare results then? 
  • missing information how many patients had x-rays and how many had MRI (line 157-158)
  • 274 - can not conclude it without comparing to KL score or MRI scans
  • you should divide groups unilateral/bilateral with the same inclusion criteria regarding x-rays or MRI scans. 
  • control (healthy or another treatment option) group missing

Author Response

Thank you for commenting on our response. 

Response to reviewer 1 (round 2)

  • bilateral group - there is n.s. difference between limb before treatment- why do You compare results then? 

A: This is correct. We outlined the baseline scores of each knee individually to show that patients in the bilateral cohort were actually suffering from the contralateral knee as well and we didn’t inject the contralateral knee just to create an additional group to analyze. We think that symptomatic bilateral OA is a different but significant entity of patients we see in clinical practice and therefore worthwhile to evaluate the effect of MFAT in this population (although small n=27). As you suggested, we removed the stats of the pre-injection comparison of the baseline scores in this population as this might be unnecessary.

  • missing information how many patients had x-rays and how many had MRI (line 157-158)

A: Baseline MRI was available in 73% (index knee) (line 280, figure 5 and table 4). Information about the remaining 27% who had an X-ray before injection was indeed missing in the manuscript and was added in the first paragraph of the result section (line 200-203). X-rays were scored for OA grade according to the Kellgren and Lawrence classification. This was added in the methodology section (line158).

  • 274 - cannot conclude it without comparing to KL score or MRI scans

A: Not sure what the reviewer means here. We state on line 274 the response rates based on the OMERACT-OARSI criteria of the bilateral cohort at 12 months post-injection. We don’t see the relation of comparing this to the radiological outcomes since the therapeutic response rate (TRR) is a pure clinical observation.

  • you should divide groups unilateral/bilateral with the same inclusion criteria regarding x-rays or MRI scans. 

A: We admit that is would be ideal to make the study methodology less complex as it seems now. Nevertheless, outcomes of the bilateral group (n=27) were clearly described as a separate study population where possible. Ideally, all subjects would have had an pre-injection MRI to simplify methodology but this was clinically not feasible. The primary objective of the study was to evaluate the clinical response rate and safety of a single MFAT infiltration in a wide range of knee OA patients. OA had to evidenced by imaging but could either be on MRI or radiographs. Since almost ¾ of the population had a pre-injection MRI, the authors considered performing a regression analysis in an attempt narrow future patient selection for MFAT therapy since these are currently absent in the MFAT literature. Nevertheless, this was considered to a be secondary objective of the study and results should therefore not be interpreted as definitive. However, the inconsistency of pre-injection imaging is a limitation and this was stated on line 358 after discussing the regression analysis for MRI parameters.

  • control (healthy or another treatment option) group missing

A: As stated in the limitation section (line 409-411), this study had merely an observational purpose of primarily identifying the response rate on MFAT in a broad knee OA population. We are aware that a control placebo group would have made this study scientifically more powerful and we certainly do not underestimate the effect of placebo/saline (50% response rate at 12 months follow-up, DOI: 10.1177/0363546517732734). Since only few clinical data are available about MFAT, we think it is more likely to test the therapy in a large population (namely for its safety) before direct comparison. In the meantime, we indeed started a prospective RCT with hyaluronic acid versus MFAT infiltration at our institute to continue on the results of the current study (line 411-413).

Reviewer 3 Report

Dear Authors, i understood your consideration but as regards the second  point i think that this is important bias to consider the wide range age.

Author Response

Thank you for commenting on our response. 
